# Weighted Gene Correlation Network Meta-Analysis Reveals Functional Candidate Genes Associated with High- and Sub-Fertile Reproductive Performance in Beef Cattle

**DOI:** 10.3390/genes11050543

**Published:** 2020-05-12

**Authors:** Pablo A. S. Fonseca, Aroa Suárez-Vega, Angela Cánovas

**Affiliations:** Centre for Genetic Improvement of Livestock, Department of Animal Biosciences, University of Guelph, Guelph, ON N1G 2W1, Canada; asuarezv@uoguelph.ca

**Keywords:** meta-analysis, RNA-sequencing, gene network, functional candidate genes, systems biology, subfertility, beef cattle

## Abstract

Improved reproductive efficiency could lead to economic benefits for the beef industry, once the intensive selection pressure has led to a decreased fertility. However, several factors limit our understanding of fertility traits, including genetic differences between populations and statistical limitations. In the present study, the RNA-sequencing data from uterine samples of high-fertile (HF) and sub-fertile (SF) animals was integrated using co-expression network meta-analysis, weighted gene correlation network analysis, identification of upstream regulators, variant calling, and network topology approaches. Using this pipeline, top hub-genes harboring fixed variants (HF × SF) were identified in differentially co-expressed gene modules (DcoExp). The functional prioritization analysis identified the genes with highest potential to be key-regulators of the DcoExp modules between HF and SF animals. Consequently, 32 functional candidate genes (10 upstream regulators and 22 top hub-genes of DcoExp modules) were identified. These genes were associated with the regulation of relevant biological processes for fertility, such as embryonic development, germ cell proliferation, and ovarian hormone regulation. Additionally, 100 candidate variants (single nucleotide polymorphisms (SNPs) and insertions and deletions (INDELs)) were identified within those genes. In the long-term, the results obtained here may help to reduce the frequency of subfertility in beef herds, reducing the associated economic losses caused by this condition.

## 1. Introduction

Intensive selection pressure has led to a decrease in fertility efficiency in both beef and dairy cattle populations [1,2]. Genetic mechanisms such as pleiotropy, genetic hitchhiking, and epistasis can be the cause of the genetic correlations, and consequent undesirable effects, observed between production and fertility traits [3,4,5,6,7]. Poor fertility and reproductive inefficiency are among the main causes of the negative impact on the profitability of both beef and dairy herds [8,9]. Embryo mortality is the major factor affecting fertility and production costs, with the majority of pregnancy losses occurring in the first month of pregnancy [10,11]. Regarding the comparison between infertility and subfertility, sub-fertile animals are more pervasive in the herds because of the fact that true infertility has a frequency of up to 5% in the herds [12,13,14]. Additionally, sub-fertile animals can generate progeny, consequently maintaining the causal alleles for this phenotype in the population [15]. On the other hand, the causal alleles for the infertile phenotype tend to reduce its allelic frequency naturally across time owing to the absence of progeny carrying the alleles. Therefore, sub-fertile animals have a higher probability to have a higher cost in the livestock industry.

Reproductive traits are considered complex phenotypes as they present a high heterogeneity, high environmental impact, and do not follow a Mendelian inheritance pattern [16]. In this sense, investigating the genes involved in complex phenotypes is not a trivial task. Especially when using high-throughput genetic tools, which usually demand a high number of samples and a high accuracy of the phenotype evaluated to obtain a consistent and significant result. RNA-sequencing technology has allowed the identification of several candidate genes and genetic variants associated with fertility traits in cattle in the last decade [17,18,19,20,21,22,23,24]. However, the majority of these studies are focused on the conventional gene by gene differential expression analysis. Other than to provide significant results to understand the genetic basis of complex traits, this approach may result in an underrepresentation of the genetic interactions between genes. The use of co-expression gene networks accounts for the expression profile across multiple samples, leading to the identification of regulatory and functional mechanisms in common [25]. The guilt-by-association (GBA) principle is one of the major measurements to evaluate the quality of co-expression networks, where genes with similar functional activities tend to have a similar expression profile, and consequently, a higher co-expression [26]. Meta-analysis approaches tend to enhance the performance of co-expression networks when the GBA principle is evaluated [27,28,29,30,31]. The application of network meta-analysis using high-throughput expression data is relatively new and can help to improve the detection of differentially expressed genes (DEGs) and to reduce the impact of differences between studies that can be hard to remove, such as bias during the library preparation step, which will implicate in spurious differences between groups [27,32]. This is reinforced by the stronger correlation observed between true log (fold-change) values and the values obtained in the network meta-analyses when these values are compared with those obtained from meta-analyses performed just merging datasets [27]. Consequently, the integration of both approaches (co-expression gene networks and network meta-analysis) can be a good alternative to increase the potential to identify functional candidate genes regulating a complex trait.

In the present work, we performed the integration of network meta-analysis and weighted gene correlation network analysis (WGCNA) approaches in order to scrutinize the co-expression and the genetic basis of high-fertile and sub-fertile phenotypes in beef cattle. Additionally, potential functional candidate variants, fixed in one of the phenotypic groups, were prospected using the RNA-sequencing data.

## 2. Materials and Methods

### 2.1. Ethics Approval and Consent to Participate

The current study integrates data from previously published studies. The respective information about the ethics approval can be found in Moraes et al., (2018) [18] and Geary et al. (2016) [20].

### 2.2. Data Collection

The RNA-sequencing (RNA-seq) data from the endometrium tissue of high- and sub-fertile beef cows (HF and SF, respectively) were retrieved from National Center for Biotechnology Information (NCBI) Gene Expression Omnibus (GEO) public database from two previously published studies: GSE81449 and GSE107891 [18,20]. In these studies, the differentially expressed genes between HF and SF beef cows were identified. A total of 20 animals (10 animals per group; HF (*n* = 10) and SF (*n* = 10) from both studies) were analyzed. Briefly, the fertility status of those animals was based on the pregnancy outcome ratio after up to four rounds of successive high-quality embryo transfer protocol of estrus synchronization (PG-6d-CIDR and GnRH), where heifers that did not exhibit standing estrus received GnRH injection on day 0. As described by Moraes et al., (2018) and Geary et al., (2016) [18,20], the pregnancy outcome was detected by ultrasound and those animals with a pregnancy success ratio equal to 100% or 25%–33% were classified as HF and SF, respectively. Additional details about the breed composition, synchronization protocol, flushes, biopsies, RNA extraction, and sequencing can be found in the original manuscripts [18,20].

### 2.3. RNA-Sequencing Data Alignment and Variant Calling

The CLC Genomics Workbench 11.0 (CLC bio, Cambridge, MA, US) was used to perform quality control (QC), read alignment, transcript quantification, and variant calling [33,34,35]. In QC, the PHRED score distribution, GC content, nucleotide contribution, and duplication levels were evaluated as described by Cánovas et al., (2014) [36]. Sequencing reads were aligned against the bovine reference genome ARS-UCD1.2 [37] using the “Map reads to reference” algorithm with the following criteria: match score = 1; mismatch cost = 2, length fraction = 0.5, and similarity fraction = 0.8. Subsequently, we quantified transcript expression (total counts) and only those genes with a fragments per kilobase of exon model per million reads mapped (FPKM) > 0.2 in both conditions (HF and SF) were maintained for the next analyses [38,39]. The variant calling was performed using the fixed ploidy variant detect algorithm (diploidy genome) on CLC Genome Workbench. A required variant probability >90%, a minimum coverage of 10, and a minimum count of 2 were set for the variant detection [34]. The base quality filter was performed using a neighborhood radius = 5, minimum central quality = 20, and minimum neighborhood quality = 15 [24]. Genetic variants (single nucleotide polymorphism, SNP; and insertion and deletion, INDEL) fixed in one of the groups were selected as potential functional variants for further analyses.

### 2.4. Identification of Genes with Expression Determined by the Study and Outliers

After filtering those genes with an FPKM > 0.2 in both conditions, the raw counts were used to perform a log-likelihood ratio test (LRT) in the DESeq2 package in R [40] in order to estimate the impact of different studies over the gene expression. Those genes with a differential expression significantly affected (adjusted false discovery rate (FDR) 5%, *p*-value < 0.05) by the different studies (GSE81449 and GSE107891), not by the conditions (HF and SF), were excluded from the analysis. Additionally, the counts for the maintained genes were used to perform a clustering analysis in order to identify potential outliers among the samples. The Manhattan distance among the animals was calculated and used in a multidimensional scaling analysis in order to cluster the animals using the first two principal components. These analyses were performed using the function cmdscale in R. The possible outliers were removed from the next steps. In this study, the outliers were classified as those animals that did not cluster following the condition (HF and SF).

### 2.5. Meta-Analysis of Differentially Expressed Genes

The DEGs were identified using the DESeq2 package in R, where a negative binomial generalized linear model was used using as a fixed effect the condition (HF and SF) of each animal [40]. Initially, this analysis was conducted for each study individually. The threshold to define a gene as DE in each dataset was maintained as described previously by Geary et al., (2016) [20] (GSE81449; adjusted *p*-value FDR 5% < 0.1 and |log(fold-change (FC))| > 2) and Moraes et al., (2018) [18] (GSE107891; adjusted *p*-value FDR 5% < 0.05 and |log(FC)| > 2). Subsequently, the netmeta package in R [41] was used to perform a network meta-analysis and calculated the combined test-statistics (*p*-values and log2(FC)) for each gene expressed in both datasets. The DEGs in the network meta-analysis approach were identified using a threshold composed by adjusted *p*-value <0.1 and |log2(FC)| >2, which is the combination of the less stringent threshold from both studies.

### 2.6. Weighted Correlation Network Analysis

Once the comparability between the datasets was confirmed, the R package WGCNA (Weighted Correlation Network Analysis) [42] was used to identify the differentially co-expressed modules (DcoEx) of genes for HF and SF groups of animals. Briefly, after QC, a soft-thresholding power was chosen based on a criterion of approximate scale-free topology. The first soft-thresholding power to reach a scale-free topology model fit ≥0.8 was selected for each group. Subsequently, the co-expression similarity matrix is raised to this soft-thresholding (8 for HF and 8 for SF) power in order to obtain the adjacency matrix. Consequently, this last matrix was used to calculate the topological overlap measure (TOM). The adjacency matrix was calculated using a signed hybrid network. The module detection was performed using the blockwiseModules function using the dynamic tree cut algorithm [43] with a minimum number of genes per module equal to 30 and the maximum size of blocks equal to 9000 (this number was selected to fit all the genes in a single module, which might increase the module detection sensibility). After the module detection by WGCNA package, the R package km2gcn [44] was used to reallocate the genes within modules using a k-means clustering approach. Finally, the final modules detected for each group were compared using the following methodology:-For each sample s in (HF and SF);-For each module m(s) in s;-Apply a Fisher’s exact test under the null hypothesis that there is no significant overlapping of m(HF) in SF and m(SF) in HF after a Bonferroni multiple test correction.

At the end of this step, the modules of genes in HF without overlapping in SF, and vice-versa, here called DcoExp modules, were selected. The interconnectivity among the genes within each DcoExp module was plotted using the igraph package [45]. In addition, the hub score of the genes within each module was estimated using the principal eigenvector of A*t(A), where A is the adjacency matrix of the module. From these results, the top 10 genes explaining the majority of the module topology were identified.

### 2.7. Functional Analysis and Annotation of Candidate Genes

The DcoExp modules with top-hub genes harboring variants fixed in one of the conditions (HF and SF) were selected for the functional analysis, here called candidate DcoExp modules. The functional analysis was conducted using the “Core Analysis” function implemented in the ingenuity pathways analysis (IPA—Ingenuity System Inc, Redwood City, CA, USA). Genes without an associated gene symbol or without gene annotation were subjected to an annotation by homology. The BioMart application [46] was used to retrieve the respective associated human gene symbol for those genes. Only the non-annotated genes, with a percentage of identity with the human homolog higher than 80%, were annotated by this approach. The enriched (*p*-value < 0.05) canonical pathways and diseases and functions for each candidate DcoExp modules were annotated. Additionally, the significant upstream regulators (FDR < 0.05 multiple testing correction) for each candidate DcoExp modules were identified. The functional candidate genes were selected among those genes within candidate DcoExp modules or in the significant upstream regulators that harbor fixed variants and are among those genes in which the expression profile was not determined by the study.

Additionally, a “guilt by association”-based prioritization approach was performed using the GUILDify and ToppGene applications on the functional candidate genes [47,48]. Initially, GUILDify was used to retrieve a “trained-list” of candidate genes associated with pre-selected phenotypes. After this step, the “trained list” obtained using GUILDify and the list of functional candidate genes are used in ToppGene. Briefly, GUILDify uses BIANA knowledge base to identify genes associated with selected phenotypes. In the present study, the phenotypes used on GUILDify were as follows: “fertility”, “fertilization”, “decidualization”, “implantation”, “preimplantation”, “endometrium”, and “embryonic development”. BIANA creates a species-specific (human, in this study) interaction network for each gene identified by GUILDify. Subsequently, a prioritization algorithm based on network topology is used to rank the genes. Genes with a GUILD score higher than 0.44 (mean + 3 × standard deviation of GUILD score) were used to create the “trained” gene list. This list was used on ToppGene and a functional annotation-based prioritization was performed using a fuzzy-based multivariate approach. ToppGene uses the functional information shared among the “trained” and the functional candidate gene lists from several sources including the gene ontology (GO) terms for the three main categories of molecular function (MF), biological process (BP), and cellular component (CC); human and mouse phenotypes; metabolic pathways; Pubmed publications; co-expression pattern; and diseases. For each functional annotation for each functional candidate gene, a *p*-value was calculated using a random sampling of 5000 genes from the whole genome. In the next step, using a statistical meta-analysis, the *p*-values were combined into a final *p*-value. Subsequently, the genes with a significant *p*-value, after an FDR of 5% multiple corrections, were selected as prioritized genes. The complete description of the meta-analysis to calculate the final *p*-value is available in Chen et al., (2009) [48]. Briefly, for each gene *G*, a *p*-value is computed by random sampling genes across the whole genome (5000 genes in this study) and a similarity score between *G* and the *functional annotation* of the trained list using different statistical approaches for categorical (e.g., GO and pathways terms) and numeric terms (expression profile in different tissues). While a fuzzy approach is applied for the categorical terms, a Pearson correlation between the expression vectors of the candidate gene and the genes in the trained list is computed. Finally, a *p*-value for each *functional annotation* of *G* is computed using a derivation from the annotation of the genes randomly sampled across the genome using the following formula:(1)$$p\left(S_{i}\right)=\frac{\left(\begin{array}{c} Number\;of\;random\;sampled\;genes\;with\;\\ similarity\;score\;higher\;than\;G\; \end{array}\right)}{\left(\begin{array}{c} Number\;of\;genes\;in\;the\;random\;sampling\;\\ processes\;whith\;functional\;annotation \end{array}\right)}$$

All the *p*-values obtained for each *G* are combined using a Fisher’s inverse chi-square method, where the *p*-values are assumed to come from an independent test:(2)$$-2\overset{n}{\underset{i=1}{\sum }}logp_{i}\to X^{2}\left(2n\right)$$

This metanalysis approach followed by multiple testing correction substantially reduces the number of false positive functional candidate genes.

## 3. Results

Figure 1 summarizes the pipeline applied in the present study to identify the functional candidate genes, and candidate genetic variants harboring those genes, within the differentially co-expressed gene modules between HF and SF animals.

### 3.1. RNA-Sequencing and Variant Calling Statistics

The number and percentage of uniquely mapped reads for each sample are shown in Appendix A. A total of 13,812 genes were expressed in both studies (GSE81449 and GSE107891) with an FPKM > 0.2. These genes were used for the DEG meta-analysis in order to estimate the combined test-statistics (*p*-value and log2(FC)) for each group (HF and SF). Additionally, independently of the DEG meta-analysis, the LRT analysis indicated that 2462 genes had the expression determined by the study, not by the condition. Consequently, 11,350 genes were used to further investigate possible outliers among the samples and initially used in the WGCA analysis.

Regarding the variant calling analysis, the genomic coordinates, type (SNP or INDEL), and functional impact for the variants fixed in one of the conditions, obtained using Ensembl Variant Effect Predictor (VeP), are shown in Appendix A. A total of 2254 variants were identified as uniquely fixed in the HF group (2113 SNPs and 141 INDELs), while 3117 variants were uniquely fixed in the SF group (3034 SNPs and 83 INDELs). The percentage of each functional class for the fixed variants in each group is shown in Appendix A.

### 3.2. Outlier Detection and Differential Expression Analysis between High-Fertile and Sub-Fertile Animals

The outlier detection analysis resulted in the exclusion of one SF animal in the GSE81449 dataset (SRR3505358) and one animal from each condition (HF: SRX3461001 and SF: SRX3461010) in the GSE107891 dataset (Appendix A). The final sample size used in the analyses was 17 animals (HF = 9 and SF = 8). The overlapping among the DEG genes was calculated for the DEG identified in the meta-analysis performed in this study, including the DEG identified using the alignment with CLC Bio Genomics against the new bovine reference genome ARS-UCD1.2 and the DEG identified originally in the previously published studies [18,20]. A very low overlap was observed across the differential expression analyses between both studies. The network meta-analysis resulted in 14 DEGs (Figure 2a, adjusted *p*-value < 0.1 and |log2(FC)| > 2). Additionally, despite the larger number of DEGs, the results obtained with CLC Bio Genomics and DESeq2 using as a reference genome for the mapping step the new bovine assembly ARS-UCD1.2 showed a small overlapping with the original result (maximum overlapping of 11 genes) (Figure 2b). The majority of DEGs in the network meta-analysis (using the non-adjusted *p*-value) are shared with the results obtained using the GSE107891 dataset when the alignment against the ARS-UCD1.2 reference genome was performed using CLC Bio Genomics. The *p*-values and log2(FC) for all the genes evaluated in the network meta-analysis, as well as the DEGs previously identified in GSE107891 and GSE81449, are presented in Appendix A.

### 3.3. Identification of Candidate Differentially Co-Expressed Gene Modules for High-Fertile and Sub-Fertile Animals

The correlation between the HF and SF gene ranked expression values (FPKM) obtained from DESeq2 package and ranked connectivity, estimated through WGCNA package, was evaluated to estimate the network conservation in the combined dataset composed of the samples from GSE107891 and GSE81449. A correlation of 0.99 (*p* < 1 × 10^−200^) was obtained for the ranked expression values and 0.4 (*p*-value < 1.7 × 10^−191^) for the ranked connectivity between the HF and SF samples. These results indicate that both datasets are suitable to be analyzed together for the identification of co-expressed gene networks owing to the strong correlation observed. Thirty-two and 34 co-expressed gene modules were identified for HF and SF, respectively. The identification of DcoEx modules (non-overlapping threshold = *p*-value < 4.59 × 10^−5^) resulted in 44 modules, 22 for each condition (Appendix A). From them, we performed an additional filtering to select the modules with at least 1 of the 10 top-hub genes harboring fixed functional candidate variants (FCVs) identified in the variant calling process (HF = 157 FCV and SF = 214 FCV). The resulting modules, called candidate DcoExp modules, were ten and two in the HF and SF datasets, respectively (Table 1). The list of enriched biological processes, diseases, and functions, as well as upstream regulators obtained for each candidate DcoExp module using IPA core module, can be found in Appendix A. It is important to highlight that, for both functional prioritization and canonical metabolic pathway enrichment analysis, we used human, mice, and rat annotations owing to the database availability and the more complete annotation status for these organisms. Consequently, it would be possible to observe small differences in the functions performed by some orthologous gene in different species. Overall, owing to the close evolutionary relationship between these species and cattle, a high level of similarities of functions performed by the orthologous genes in those species is expected.

### 3.4. Functional Candidate Genes

The functional candidate genes, top-hub genes from the candidate DcoExp modules, and upstream regulators of the DcoExp modules identified with the core analysis from IPA software, harboring FCV, are shown in Table 2 and Table 3. This candidate gene list is composed of a total of 52 genes: 24 top-hub genes from the candidate DcoExp modules harboring FCV, and 28 upstream regulators harboring FCV. Additionally, functional candidate genes were using a “guilt by association”-based prioritization analysis (ToppGene sofware) using a trained dataset of genes related to fertility obtained from GUILDify. The adjusted (FDR) overall *p*-value of the significantly prioritized candidate genes from the functional prioritization analysis is shown in Table 2. It is important to highlight that the top hub-gene from Cyan module (ENSBTAG00000046047) with a fixed candidate variant was not analyzed using ToppGene owing to the lack of gene symbol annotation (even after the annotation by homology process). The functional prioritization resulted in 32 significantly prioritized functional candidate genes. The enriched terms for the traied dataset and the complete prioritization result for the candidate genes are provided in Appendix A. The FCV identified within the genomic coordinates of the prioritized genes and the corresponding functional consequence is shown in Appendix A. The relationship between the functional candidate genes and the candidate DcoExp modules is shown in Figure 3. The turquoise HF module showed the highest number of related prioritized functional candidate genes (14), while the pale turquoise SF module showed the smallest number of related genes (1). The percentage of each functional consequence and the number of fixed variants per each prioritized functional candidate gene is shown in Figure 4. In total, 100 FCV were identified in the 32 significantly prioritized functional candidate genes.

## 4. Discussion

### 4.1. Network Meta-Analysis for Identification of Differentially Expressed Genes between High-Fertile and Sub-Fertile Animals

The integration of multiple datasets and genetic information from different levels has been shown to be a powerful strategy for the identification of candidate genes in livestock [3,49,50,51]. The network meta-analysis performed in the present study reinforced the association between the expression profile of some genes with the high- or sub-fertile condition. The 14 DEGs identified by the network meta-analysis (adjusted *p*-value < 0.1 and |log(fold-change| > 2) were not identified as DE in the original results of both previous studies (GSE107891 and GSE81449) described by [18,20]. This result can be explained by the differences in the assemblies, alignment, and quantification algorithms, as well as the detection power improvement observed in the meta-analysis performed in the present study. However, 11 genes were shared with the GSE107891 dataset using the new bovine reference genome ARS-UCD1.2 by CLC BIO genomics. In general, a small overlap was observed in all of the comparison scenarios (Figure 2b). The Simpson’s paradox is a common phenomenon observed in biological analysis that can help to address these differences across the analyses. Briefly, the Simpson’s paradox occurs when results from combined datasets contradict those from the individual analysis. The impact of Simpson’s paradox was already discussed in several fields, such as gene expression network analysis [52,53,54]. Despite biological bases for the Simpson’s paradox still being poorly understood, there are some points that must be highlighted in the present study. First, despite both studies analyzing the transcriptome of the same tissue (endometrium), GSE81449 analyzed a dataset obtained from endometrial biopsy from day 14 post-estrus, while GSE107891 analyzed a dataset from day 17 post-estrus. This difference, together with the population effect, the new mapping strategy (different software and bovine reference genome), and the software used for DE analysis, can affect the transcriptional profile of each sample, consequently resulting in a strong study-dependent effect (as observed in the LRT analysis). Additionally, as shown in Appendix A, few genes were identified as DE between HF and SF animals in the original results of both studies, with the majority of these DEGs composed by non-annotated genes (LOCs) or genes with poorly understood biological function. Here, a larger number of DEGs were obtained in the individual datasets. However, the network meta-analysis resulted in a similar number of DEGs, but used a non-adjusted *p*-value threshold. These results reinforce the difficulty in validating and identifying functional candidate genes using the traditional gene by gene differential expression analysis when complex traits are analyzed without the comparison of extreme groups, such as HF and SF cows. For example, Moraes et al. (2018) [18] identified a significantly larger number of DEGs when comparing high-fertile versus infertile animals. Consequently, new strategies must be applied to better understand the genetic differences between HF and SF animals.

### 4.2. Differentially Co-Expressed Modules and the Identification of Functional Candidate Genes 

The combination of co-expression gene networks, the identification of top hub-genes, the identification of fixed genetic variants in HF or SF group of cows, and the functional analysis using the *p*-values and log2(FC) obtained in the network meta-analysis were used to prospect for functional candidate genes regulating the differences between fertile groups. The integration of different sources of biological information is a powerful tool for the identification of functional candidate genes, which has already resulted in interesting results in livestock species [3,49,50,51,55,56]. Here, in order to avoid a massive discussion about all the results obtained, only the main achievements will be addressed. Thirty-two prioritized functional candidate genes (22 upstream regulators and 10 top hub-genes), related to 11 candidate DcoExp modules, were identified.

The upstream regulators genes of candidate DcoExp modules, harboring functional candidate variants, were follows: *DLG1* (Tan HF), *AGER* (Saddlebrown HF), *SORT1* (Saddlebrown HF), *HNRNPAB* (Saddlebrown HF), *TBX6* (Red HF), *HSF1* (Purple HF and Darkgreen HF), *LEPR* (Lightgreen HF), *DYSF* (Lightgreen HF), *BCKDK* (Lightgreen HF), *CHFR* (Grey60 HF), *ERN2* (Green HF), and *RDH10* (Darkgreen HF). Among these genes, it was possible to identify relevant biological processes associated with fertility, such as oocyte polarization during maturation (*DLG1*), disorders of the müllerian ducts (*TBX6*), regulation of affects gonads or gonadotrophs (*LEPR*), and reproductive success (*HSF1*) [57,58,59,60,61,62].

In the HF group, the turquoise module showed the largest number of related prioritized functional candidate genes, 14 in total. This module is enriched for relevant biological processes (FDR < 0.05), such as regulation of inositol metabolism (3-phosphoinositide degradation, superpathway of inositol phosphate compounds, D-myo-inositol (1,4,5,6)-tetrakisphosphate biosynthesis, and so on) and estrogen-mediated S-phase entry. The inositol metabolism and the cell cycle mediated by estrogen activity are very important signaling pathways associated with the cellular proliferation in the uterus [63]. Genes located within these modules such as *DUSP16*, *DUSP2*, and *PIK3AP1* are directly associated with the regulation of phosphatidylinositol activity [64,65].

The top hub-genes from turquoise HF module, harboring fixed variants in HF or SF animals, and prioritized in the functional analysis, were *ARHGEF16*, *IFT80*, and *PIGR*. Rho Guanine nucleotide exchange factor 16 (*ARHGEF16*) codifies an ELMO1 interacting protein responsible to promote the clearance of apoptotic cells in a RhoG-dependent and Dock1-independent manner [66]. The relationship between *ARHGEF16* and fertility has yet to be described in the literature, such as by Elliott et al. (2010) and Gong et al. (2018). Interestingly, ELMO1 knockout mice presented multinucleated giant cells, uncleared apoptotic germ cells, and decreased sperm output in Sertoli cells owing to the phagocytic deficiency [67,68]. However, there is no link between Elmo1 and female fertility currently described. The intraflagellar transport 80 (*IFT80*) codifies a intraflagellar transport protein responsible for regulating the Jeune asphyxiating thoracic dystrophy, osteoblast, and chondrocyte differentiation [69,70,71]. Despite the absence of a direct link between *IFT80* and fertility, the regulation of osteoblast and chondrocyte is essential for embryo survival [72,73]. Interestingly, *VEGFA*, another gene located within the turquoise HF module, is crucial for chondrocyte survival and bone development in the embryonic stage [74]. The polymeric immunoglobulin receptor (*PIGR*) encodes a poly-g receptor in epithelial cells responsible for controlling the transcytosis process that can be regulated by steroids, such as estrogen [75,76]. In uterine epithelial cells, *PIGR* is responsible for transporting polymeric IgA. In rats, the transcriptional levels of *PIGR* are higher in the estrous when compared with proestrus or diestrus [77]. Variations in the immunoglobulin diversity and quantity in the uterus are observed during the ovulatory process, indicating a key regulatory role of ovarian hormones. Consequently, this suggests there is an impact on fertility [78,79]. Additionally, proteomic analysis of fertile and sub-fertile hens suggested that the levels of *PIGR* decreased 24 h after insemination in the uterine fluid, with the main location in the uterovaginal sperm storage tubules (SST), suggesting a response caused by the sperm arrival. These results suggest that *PIGR* responds to sperm arrival in both scenarios in sub-fertile hens, which could be a result of the higher transport activity of IgA and secretory complex to the lumen of SST [80]. It is important to highlight that *PIGR* was the second candidate gene with the highest number of fixed functional candidate variants. All the variants were identified as fixed in HF animals. Five missense variants were identified as fixed in the *PIGR* gene, where four were previously described (rs41790811, rs41790822, rs41790826, and rs41580873) and one is a new variant (c.627A > C or p.Ile162Arg). Two of these variants (rs41790822 and rs41790826) have a predicted deleterious effect based on the Sorting Intolerant from Tolerant (SIFT) score (0.04 and 0.05, respectively). The identification of missense variants with predicted deleterious effect in the HF animals might corroborate the hypothesis that higher activity levels of *PIGR* are associated with the sub-fertile phenotype in hens, as proposed by Riou et al. (2019) [80].

The significant upstream regulators genes identified by the IPA core analysis associated with the turquoise HF module and harboring fixed functional candidate variants were *PGR*, *EGFR*, *PIK3C2A*, *CUX1*, *TCHP*, *ETV5*, *KLF4*, and *ARID4A*. The progesterone receptor (*PGR*) is crucial for the initiation of pregnancy and subsequent preservation of uterus health [81]. Consequently, *PGR* is an interesting marker for uterine receptivity during implantation [82]. In this study, three fixed SNPs in HF animals were identified in the downstream region of *PGR*. Two of those SNPs were already described (rs208289597 and rs208479533) and one is new (g.- 1458T > C). It is important to highlight that the *PGR* was identified as an upstream regulator of the turquoise module, however, it is also one of the genes that composes this module. Additionally, *PGR* was also identified as an upstream regulator of the cyan HF module. Epidermal growth factor receptor (*EGFR*) is a member of the epidermal growth factor family that plays crucial roles in the regulation of female fertility [83]. Among its functions, *EGFR* regulates puberty, oocyte maturation, uterine development, embryo implantation, and placental overgrowth [83]. Phosphatidylinositol-4-phosphate 3-kinase catalytic subunit type 2 α (*PIK3C2A*) is a member of the PI3-kinases acting in cell proliferation, oncogenic transformation, cell survival, cell migration, and intracellular protein trafficking [84]. The phosphatidylinositol 3 kinase (PI3K) pathway plays a crucial role in the control of mammalian oocyte growth and early follicular development [85]. Cut like homeobox 1 (*CUX1*) codifies a DNA binding protein related to the control of cell cycle progression, cell motility, and invasion [86]. Recently, SNPs on *CUX1* were associated with cow and heifer conception rate in Holstein cattle [87]. Trichoplein keratin filament binding (*TCHP*) is associated with cytoskeleton remodeling neurofilaments and vulvar sarcoma according to MalaCards database (ID: VLV038). However, there is no direct evidence linking this gene with fertility status. ETS variant 5 (*ETV5*) is a transcription factor that plays crucial roles in male fertility, acting in the spermatogonial stem cell self-renewal and maintenance of spermatogonial stem cell niche [88,89,90]. Female knockout mice for *ETV5* are infertile owing to a decreased ovulation and no interest in mating [91]. Kruppel like factor 4 (*KLF4*) is a transcription factor required for normal development of the barrier function of skin and with the ground state of pluripotent stem cells [92]. Regarding female fertility, *KLF4* mediates the anti-proliferative effects of progesterone during the G0/G1 arrest in endometrial epithelial cells [93]. AT-rich interaction domain 4A (*ARID4A*) is a nuclear binding protein that acts as a transcriptional coactivator of androgen receptor and retinoblastoma-binding protein during the regulation of Sertoli cell function, consequently playing a crucial role in male fertility [94]. However, disregarding the impact of target therapies for ovarian and endometrial cancer, there is no direct link between *ARID4A* and female fertility [95]. Interestingly, dual specificity phosphatase 16 (*DUSP16*) and prominin 1 (*PROM1*) are genes within the turquoise HF modules that act like top hub-genes in other candidate DcoExp modules (ligthgreen HF and green HF, respectively). The function of *DUSP16* was described previously, while *PROM1* is a pentaspan transmembrane glycoprotein that was already identified as DE during the window implantation [96]. It is important to highlight that, even without a direct discussion addressed, all these genes harbour fixed variants identified in HF or SF animals.

The other top hub-genes of candidate DcoExp modules in the HF group, harboring fixed functional variants, were as follows: *IQCG* (Tan H), *MAPKAP1* (Grey60 HF), *MDM4* (Red HF), *F2RL2* (Saddlebrown HF), *MIA3* (Green HF), and *PPP1R12B* (Lightgreen HF). IQ motif containing G (*IQCG*) was the gene with the largest number of fixed variants (14 variants, all mapped in the upstream region of the gene and fixed in HF animals). IQ motif containing G is a key regulator of ciliary/flagellar motility that plays a crucial role in the formation of the sperm flagellum and spermiogenesis in mice [97,98]. However, to the best of our knowledge, there is no direct link between female fertility and *IQCG*. The MAPK associated protein 1 (*MAPKAP1*) is a member of FRAP1 complex. The disruption of the FRAP1 complex leads to post implantation lethality caused by the impaired cell proliferation and hypertrophy of embryonic disc and trophoblast [99]. The *MAPKAP1* gene is one of the top hub-genes of the Grey60 HF module, which is enriched for several fertility related biological processes, such as cell cycle regulation, PI3K signaling pathway, and triacylglycerol metabolism. MDM4 regulator of P53 (*MDM4*) encodes a protein responsible for inhibiting p53 function. Regarding fertility, *MDM4* variants are associated with the susceptibility of ovarian and endometrial cancer [100]. Coagulation factor II thrombin receptor like 2 (*F2RL2*) is a transmembrane G protein-coupled cell surface receptor that is differentially expressed (down-regulated) during pregnancy [101]. Additionally, the Saddlebrown HF module is enriched for fertility-related processes such as androgen signaling, sperm motility, and estrogen-dependent breast cancer signaling. To the best of our knowledge, MIA SH3 domain ER export factor 3 (*MIA3*) and protein phosphatase 1 regulatory subunit 12B (*PPP1R12B*) do not have a direct link with fertility.

Interestingly, the only candidate DcoExp module identified in the SF animals maintained after the functional analysis was the pale turquoise SF. The top-hub gene identified in this module harboring functional candidate variants was *IFT80*. This gene was also identified as a hub-gene in the turquoise HF module and its association with fertility was mentioned above. The fixed variant identified in *IFT80* is mapped in a splice donor site and it was fixed in the SF animals. The analysis of the enriched canonical pathways, diseases, and functions enriched for this module highlighted a specialization for cell cycle, embryonic development, and cell death and survival. Additionally, an interesting overlap between the canonical pathways enriched in the turquoise HF and the pale turquoise SF is observed regarding the metabolism of inositol, which, as described before, is related to fertility status. The pale turquoise SF was also enriched for ketogenesis and ketolysis (FDR < 0.05), which are relevant processes in animals subjected to a high selective pressure for production traits. Within this module, 3-hydroxybutyrate dehydrogenase 2 (*BDH2*) is the main gene associated with ketone body metabolism [102,103]. Interestingly, a polymorphism in *BDH2* was associated with days open and services per conception in Holstein cows, reinforcing a possible role of this gene with fertility traits in cattle [104].

## 5. Conclusions

The results obtained here reinforce the increase in detection power for functional candidate genes in the context of data integration. The approach applied in the present study, combining co-expression gene networks, identification of top hub-genes, identification of fixed genetic variants in HF or SF cows, and functional analyses, resulted in the identification of highly relevant functional gene networks and candidate genes associated with fertility status in beef cattle. These results contribute to the better understanding of the genetic architecture of high- and sub-fertility cows. Additionally, candidate functional variants were identified uniquely fixed in one of the fertility groups (HF or SF animals) in the functional prioritized genes. Consequently, these variants could be used in genetic selection programs to validate the contribution of these variants in the fertility status. In the long-term, the results obtained here may help to reduce the frequency of subfertility in beef herds, helping to reduce the economic losses caused by this condition.

## Figures and Tables

**Figure 1 genes-11-00543-f001:**
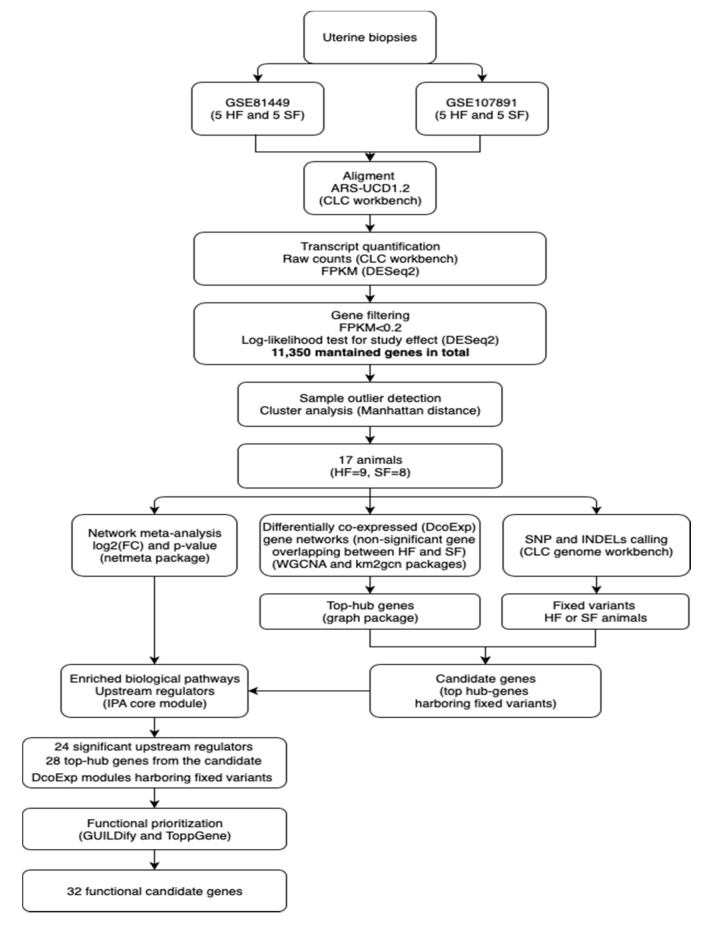
Pipeline for identification of functional candidate genes regulating differentially co-expressed gene networks between high-fertile and sub-fertile beef cows. SF, sub-fertile; HF, high-fertile; INDEL, insertion and deletion; SNP, single nucleotide polymorphism; FPKM, fragments per kilobase of exon model per million reads mapped; WGCNA, weighted gene correlation network analysis; IPA, ingenuity pathways analysis; FC, fold-change.

**Figure 2 genes-11-00543-f002:**
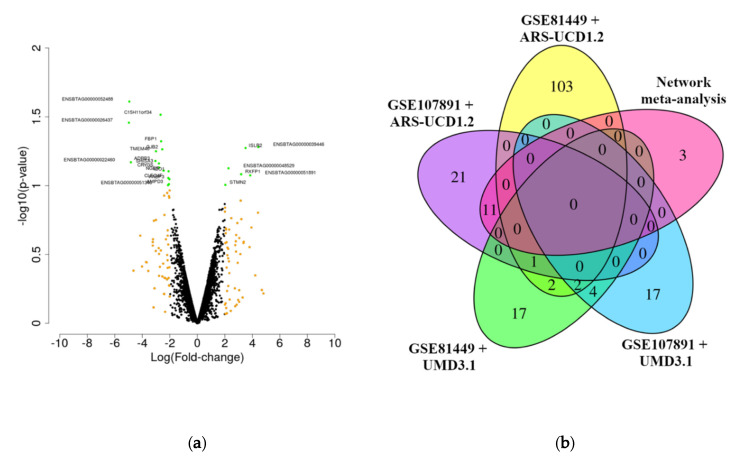
Volcano plot for the differentially expressed genes identified in the network meta-analysis (non-adjusted *p*-value < 0.1 and |log(fold-change)| > 2) (**a**) and Venn diagram comparing the results obtained in the different datasets of differentially expressed genes identified using different bovine reference genomes assemblies (**b**). In (**a**), the green dots represent differentially expressed genes identified using non-adjusted *p*-value < 0.1 and |log(fold-change)| > 2. The yellow dots represent differentially expressed genes identified using a |log(fold-change)| > 2. In (**b**), the differentially expressed genes identified in each dataset are represented in red (network meta-analysis), yellow (GSE107891 data set using ARS-UCD1.2 bovine reference genome), purple (GSE81449 data set using ARS-UCD1.2 bovine reference genome), dark green (GSE107891 data set using UMD 3.1 bovine reference genome; Moraes et al. (2018)), and blue (GSE81449 data set using UMD 3.1 bovine reference genome; Geary et al. (2016)).

**Figure 3 genes-11-00543-f003:**
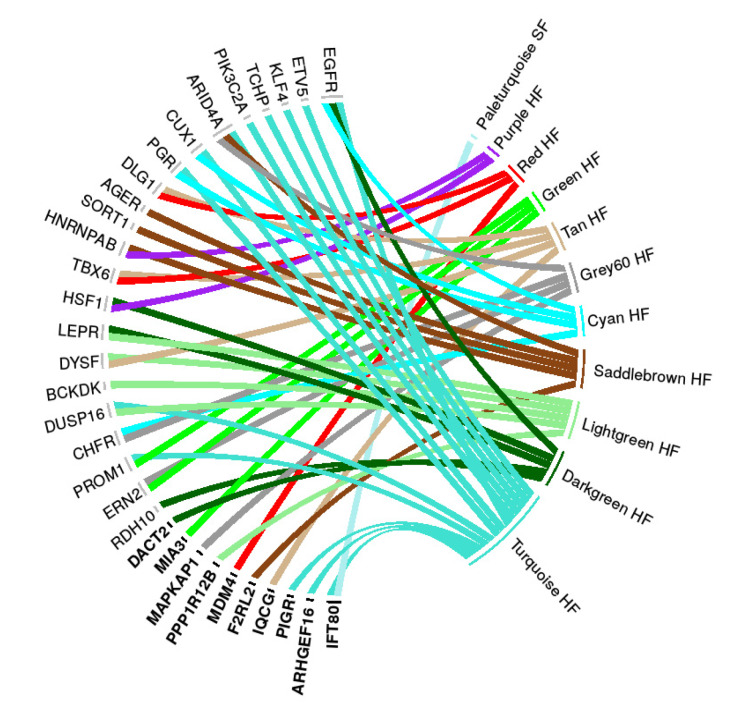
Relationship between the prioritized functional candidate genes (left-hand side) and the candidate differentially co-expressed modules (DcoExp, right-hand side). The genes in bold are the top hub-genes from the DcoExp modules, while the other genes are the significant upstream regulators genes. On the right-hand side, the HF modules are those identified in the high-fertile animals and the SF modules are those identified in the sub-fertile animals.

**Figure 4 genes-11-00543-f004:**
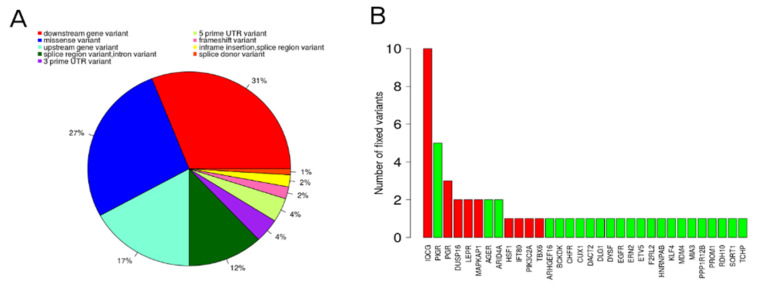
Percentage of each functional consequence annotated (**A**) and number of fixed variants per each functional candidate gene (**B**). The red bars in B represent genes where the fixed variants were identified in sub-fertile animals, while the green bars represent those variants fixed in high-fertile animals.

**Table 1 genes-11-00543-t001:** Differentially co-expressed gene modules (DcoExp), and their respective top 10 hub-genes between high-fertile (HF) and sub-fertile (SF) animals.

Module	Number of Genes	Top Hub-Genes
Cyan HF	204	ANKRD65, TIMM17A, **ENSBTAG00000046047**, ENSBTAG00000051586, RRP1, PWP2, PSMB5, LTBP2, C11H2orf81, **ENSBTAG00000050675**
Darkgreen HF	147	SREBF2, **DACT2**, SDHA, PPFIA4, ENSBTAG00000047824, ENSBTAG00000052047, ELF3, ENSBTAG00000051421, NPTN, DHRS4
Grey60 HF	177	IARS2, ENSBTAG00000053801, **ENSBTAG00000033740**, ENSBTAG00000048975, WRB, ENSBTAG00000052845, FAM214A, EIF2AK3, MPV17, **MAPKAP1**
Lightgreen HF	136	CEP104, PKP1, **PPP1R12B**, ENSBTAG00000051541, ENSBTAG00000049133, ARF6, NUMB, **SLC25A15**, **EEF1AKMT1**, **PARP4**
Purple HF	227	TIRAP, PYCR2, **FMO2**, MIIP, ENSBTAG00000049485, ENSBTAG00000054279, ENSBTAG00000052750, EAF1, SDR39U1, TINF2
Red HF	237	STRADA, ENSBTAG00000054600, **MDM4**, MARK1, KLHL20, CACYBP, ABL2, **RABIF**, ENSBTAG00000051120, TMEM50B
Saddlebrown HF	114	NME7, **CCDC181**, ENSBTAG00000054228, TCTEX1D2, IL20RB, ITGB2, ENSBTAG00000023186, **F2RL2**, OIP5, DUT
Tan HF	172	ZMAT2, CPOX, **IQCG**, WDR53, ENSBTAG00000042475, **HACL1**, INTS14, ENSBTAG00000043377, REC8, RPS27L
Turquoise HF	544	SLC45A3, IKBKE, **PIGR**, PRRX1, TNFRSF1B, SPSB1, CAMTA1, NOL9, TNFRSF25, **ARHGEF16**
Green HF	240	CTCF, **MIA3**, PSEN2, CASZ1, SDF4, **COLGALT2**, ENSBTAG00000054874, ENSBTAG00000051836, ENSBTAG00000051084, GOLGB1
Lightgreen SF	244	SOX13, TMEM81, COQ8A, ZBTB48, VWA1, EFHD2, **PWP2**, PLEKHO2, **NRDE2**, MALL
Paleturqouise SF	130	MTHFR, **ENSBTAG00000031572**, JDP2, CCDC142, CD8A, DNAJC27, ENSBTAG00000053045, ENSBTAG00000048432, CABLES2, NECAB3

Note: In bold, those genes harboring functional candidate variants are highlighted.

**Table 2 genes-11-00543-t002:** Prioritization result for top hub-genes, harboring functional candidate variants, from differentially co-expressed modules between high-fertile (HF) and sub-fertile (SF) animals.

Gene ID	Mapped Candidate Modules	Top Hub-Gene Candidate Modules	Overall Adjusted *p*-Value (FDR 5%)
ENSBTAG00000046047	Cyan HF	Cyan HF	NA *
PWP2	Cyan HF, Lightgreen SF	Cyan HF, Lightgreen SF	0.112
DACT2	Darkgreen HF	Darkgreen HF	0.028 **
MIA3	Green HF	Green HF	0.032 **
COLGALT2	Green HF	Green HF	0.109
SKA2	Grey60 HF	Grey60 HF	0.05
MAPKAP1	Grey60 HF	Grey60 HF	0.018 **
PPP1R12B	Lightgreen HF	Lightgreen HF	0.026 **
SLC25A15	Lightgreen HF	Lightgreen HF	0.09
EEF1AKMT1	Lightgreen HF	Lightgreen HF	0.1
PARP4	Lightgreen HF	Lightgreen HF	0.052
FMO2	Purple HF	Purple HF	0.061
MDM4	Red HF	Red HF	0.018 **
RABIF	Red HF	Red HF	0.109
CCDC181	Saddlebrown HF	Saddlebrown HF	0.085
F2RL2	Saddlebrown HF	Saddlebrown HF	0.028 **
IQCG	Tan HF	Tan HF	0.022 **
HACL1	Tan HF	Tan HF	0.09
PIGR	Turquoise HF	Turquoise HF	0.026 **
ARHGEF16	Turquoise HF	Turquoise HF	0.028 **
NRDE2	Lightgreen SF	Lightgreen SF	0.278
IFT80	Turquoise HF, Paleturquoise SF	Paleturquoise SF	0.028 **

* NA = not applicable as it was not possible to obtain a gene symbol for this transcript; ** significant prioritization at a significance level of 0.05 after false discovery rate (FDR) correction.

**Table 3 genes-11-00543-t003:** Prioritization result for the significant upstream regulators, harboring functional candidate variants, from differentially co-expressed modules between high-fertile (HF) and sub-fertile (SF) animals.

Gene ID	Mapped Candidate Modules	Regulated Module	Target Genes	Adjusted *p*-Value (FDR 5%) Upstream Regulation	Overall Adjusted *p*-Value (FDR 5%) for Prioritization
EGFR	Darkgreen HF	Turquoise HF	BIRC5, CCNA2, CXCL5, E2F1, EXOSC5, FKBP11, FOXP3, GFAP, HMGB3, HNRNPA1, IGBP1, ITGA6, MYBL2, PDK1, PROM1, PSEN1, RANBP1, SEMA7A, SKP2, TUBA4A, TUBB4A, VEGFA	0.003	0.002 *
EGFR	Darkgreen HF	Cyan HF	CCT5, EIF5A, EPS15, GADD45A, NUTF2, ODC1, PPIA, PSMB5, STAT3, TPST1	0.003	0.002 *
ETV5	-	Turquoise HF	AQP5, CHSY1, KRT19, KRT7, MYB, RAB27A, TJP3, VEGFA	0.016	0.009 *
KLF4	-	Turquoise HF	CCND2, CRABP2, DUSP5, E2F1, HES1, KRT14, KRT19, KRT7, MSX2, PAX2, PROM1, VEGFA, WNT5A	0.032	0.006 *
TCHP	-	Turquoise HF	VEGFA	0.039	0.028 *
COX7A2	-	Turquoise HF	STAR	0.039	0.147
PIK3C2A	-	Turquoise HF	VEGFA	0.039	0.009 *
ARID4A	-	Turquoise HF	E2F1, FOXP3	0.039	0.046 *
ARID4A	-	Saddlebrown HF	HOXB6	0.039	0.046 *
ARID4A	-	Grey60 HF	HOXB3, HOXB5	0.039	0.046 *
CUX1	Cyan HF	Turquoise HF	CCNA2, LTF, RAB36, WNT5A	0.047	0.006 *
PGR	Turquoise HF	Turquoise HF	AK3, HES1, HPGD, ITGA6, MSX2, NPC1, PDGFA, PGR, PPM1H, PRRX1, TAT	0.047	0.006 *
PGR	Turquoise HF	Cyan HF	LIG1, MAP2K3, STAT3, TSC22D3, UCK2, URB2	0.047	0.006 *
IPO9	-	Turquoise HF	PTK2B	0.047	0.09
DLG1	Red HF	Tan HF	KCNJ2	0.045	0.006 *
AGER	-	Saddlebrown HF	CCL4, TJP1	0.046	0.006 *
SORT1	-	Saddlebrown HF	UBE2I	0.047	0.009 *
HNRNPAB	Purple HF	Saddlebrown HF	TJP1	0.047	0.031 *
TBX6	Tan HF	Red HF	HES7	0.049	0.024 *
HSF1	-	Purple HF	CCT4, FKBP4, HSF2, HSP90AA1, HSPA8, HSPH1, KNTC1, RELA, SPHK2, STIP1	0.003	0.009 *
HSF1	-	Darkgreen HF	CSRP2, EFEMP1, INHBB, RPL22	0.003	0.009 *
NUB1	-	Red HF	NEDD8	0.046	0.088
DPH5	-	Red HF	NFKBIA, RELA	0.046	0.077
LEPR	Darkgreen HF	Lightgreen HF	ANGPTL4, CDK2, MMP7, PLP1, SOCS2	0.044	0.006 *
DYSF	Tan HF	Lightgreen HF	CD48, DNAJB1, FCGR2B	0.044	0.012 *
API5	Purple HF	Lightgreen HF	CDK2	0.044	0.077
BCKDK	-	Lightgreen HF	PLP1	0.047	0.041 *
DUSP16	Turquoise HF	Lightgreen HF	VCAM1	0.047	0.014 *
CHFR	Cyan HF	Grey60 HF	PLK1	0.046	0.032 *
PROM1	Turquoise HF	Green HF	DSG2	0.044	0.006 *
ERN2	Grey60 HF	Green HF	XBP1	0.046	0.031 *
UTP3	-	Darkgreen HF	IGLL1/IGLL5	0.043	0.112
RDH10	-	Darkgreen HF	RDH5	0.045	0.032 *

* Significant prioritization at a significance level of 0.05 after false discovery rate correction.

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
