# Peer review of "Weighted Gene Correlation Network Meta-Analysis Reveals Functional Candidate Genes Associated with High- and Sub-Fertile Reproductive Performance in Beef Cattle"

_genes, 2020, doi:10.3390/genes11050543_

Round 1
Reviewer 1 Report
The study is interesting and opens to a new combined approach to investigate gene networks and candidate genes. The analysis is well performed, especially the discussion which is well elaborated. I have found some errors in the manuscript related to citations: the style must be uniform (e.g. L76, L102), in addition, please remove comment (i.e. L111 and L120) and add authors name followed by the number of citation (L85, L94, L122, L188, L94 (Discussion) etc.). In Discussion, some citation should be added, namely at L66 and L101.
Please remove bold when you cite Figure and Table in the text, leave it only for the captions. Also L301 and L302 are in bold.
In L209 add abbreviations for high fertility and sub fertility.
In the title of Table 3, please remove the significance level, because you have explained it at the end of the table (right position).
Some genes are not in capital (e.g. L69 - Discussion).
Please, control References because in some citations pages and volumes missed. Moreover, authors have to check all manuscript: double spaces are present.
Author Response
Reviewer 1
The study is interesting and opens to a new combined approach to investigate gene networks and candidate genes. The analysis is well performed, especially the discussion which is well elaborated. I have found some errors in the manuscript related to citations: the style must be uniform (e.g. L76, L102), in addition, please remove comment (i.e. L111 and L120) and add authors name followed by the number of citation (L85, L94, L122, L188, L94 (Discussion) etc.). In Discussion, some citation should be added, namely at L66 and L101.
Thank you for all the valuable comments and suggestions. We addressed all the points raised during the review. The modifications are highlighted in yellow in the current version for the manuscript. We are addressed the individual questions bellow.
Please remove bold when you cite Figure and Table in the text, leave it only for the captions. Also L301 and L302 are in bold.
Answer: Thank you for the comments. In the current version of the manuscript, we removed all the bold when we cited the Figures and Tables across the manuscript.
In L209 add abbreviations for high fertility and sub fertility.
Answer: Thank you for the comments. The definitions for high fertility and sub fertility (HF and SF, respectively) were presented on line 79 in the current version of the manuscript.
In the title of Table 3, please remove the significance level, because you have explained it at the end of the table (right position).
Answer: Thank you for the comments, we removed the significance level from the title.
Some genes are not in capital (e.g. L69 - Discussion).
Answer: Thank you for the comments, in the case of “elmo1” we must represent the gene symbol in lower case as we are referring to the mouse transcript. By definition, the gene symbols in mouse are represented in lower case.
Please, control References because in some citations pages and volumes missed. Moreover, authors have to check all manuscript: double spaces are present.
Answer: Thank you for the comments, all the references were reviewed and formatted according to the journal guidelines. We also checked all the manuscript searching for double spaces and typos.
Reviewer 2 Report
Fonseca et al., using a meta-analysis approach tried to revealed functional candidate genes associated with high- and sub-fertile reproductive performance in beef cattle. A very interesting approach especially for further implementation or test of genetic improvement designs.
The manuscript is in a very good stage for publication. Some minor topics that will improve the level of publication I believe is of utmost importance to be corrected or clarified
GUILDify section: analysis based on human selected specie but the manuscript is refered to bovine. Why? Please highlight any limitation of such approach
line 10: may-->could
line 60: batch effect? hard to understand
line 115: cmdscale ()--> empty brackets?
line 204-204-->correct typos of fonts
p.17 line 164-165: hard to understand
Reference section: please correct part of this section according to journal's prerequisite, in order all references to have the same format.
Author Response
Reviewer 2
Fonseca et al., using a meta-analysis approach tried to revealed functional candidate genes associated with high- and sub-fertile reproductive performance in beef cattle. A very interesting approach especially for further implementation or test of genetic improvement designs.
The manuscript is in a very good stage for publication. Some minor topics that will improve the level of publication I believe is of utmost importance to be corrected or clarified
Thank you for all the valuable comments and suggestions. We addressed all the points raised during the review. The modifications are highlighted in yellow in the current version of the manuscript. We are addressed the individual questions bellow.
GUILDify section: analysis based on human selected specie but the manuscript is refered to bovine. Why? Please highlight any limitation of such approach
Answer: Thank you for the suggestion. Regarding the GUILDify analysis, we use human data because is the most deeply annotated database available on the web server (H. sapiens, M. musculus, Rattus novergicus, S. cerevisiae, C. elegans, D. melanogaster, and A. thaliana). Additionally, among the available organisms, the H. sapiens is the evolutionary closest specie to cattle. Based on this evolutionary proximity and the level of information available, the use of human information may be extremely useful. Indeed, use human annotation on livestock functional studies is a common procedure:
Fonseca, P. A. S., dos Santos, F. C., Lam, S., Suárez-Vega, A., Miglior, F., Schenkel, F. S., ... & Cánovas, A. (2018). Genetic mechanisms underlying spermatic and testicular traits within and among cattle breeds: systematic review and prioritization of GWAS results. Journal of animal science, 96(12), 4978-4999.
Kominakis, A., Hager-Theodorides, A. L., Zoidis, E., Saridaki, A., Antonakos, G., & Tsiamis, G. (2017). Combined GWAS and ‘guilt by association’-based prioritization analysis identifies functional candidate genes for body size in sheep. Genetics Selection Evolution, 49(1), 41.
Moraes, J. G., Behura, S. K., Geary, T. W., Hansen, P. J., Neibergs, H. L., & Spencer, T. E. (2018). Uterine influences on conceptus development in fertility-classified animals. Proceedings of the National Academy of Sciences, 115(8), E1749-E1758.
Tarsani, E., Kranis, A., Maniatis, G., & Kominakis, A. (2018). Investigating the functional role of 1,012 candidate genes identified by a Genome Wide Association Study for body weight in broilers. Proc. World Congr. Genet. Appl. to Livest. Prod. Species-Avian, 1, 564.
Morota, G., Peñagaricano, F., Petersen, J. L., Ciobanu, D. C., Tsuyuzaki, K., & Nikaido, I. (2015). An application of Me SH enrichment analysis in livestock. Animal genetics, 46(4), 381-387.
The main potential limitation of this approach is a possible change of function of an orthologous gene during the evolutionary processes, which might cause a confusion during the functional interpretation. We added this consideration on lines 271-276.
line 10: may-->could
Answer: Thank you for the suggestion, we replaced may by could in this version of the manuscript.
line 60: batch effect? hard to understand
Answer: Thank you for the comments. In this version of the manuscript we replace “batch effect” to “differences”. See Line 60.
line 115: cmdscale ()--> empty brackets?
Answer: Thank you for the comments. The empty brackets were used to represent the R functions. However, in order to avoid confusions, we removed the empty brackets in this current version.
line 204-204-->correct typos of fonts
Answer: Thank you for the comments. We reviewed the fonts across all the manuscript and standardized it.
p.17 line 164-165: hard to understand
Answer: Thank you for the comment. In the current version of the manuscript we rewrote these sentences in order to make it easier to understand (Lines 164-168): “The pale turquoise SF was also enriched for ketogenesis and ketolysis (FDR<0.05), which are relevant processes in animals subjected to a high selective pressure for production traits. Within this module 3-Hydroxybutyrate Dehydrogenase 2 (BDH2) is the main gene associated to ketone body metabolism [103,104]. Interestingly, a polymorphism in BDH2 was associated with days open and services per conception in Holstein cows, reinforcing a possible role of this gene with fertility traits in cattle [105].”
Reference section: please correct part of this section according to journal's prerequisite, in order all references to have the same format.
Answer: Thank you for the comments, all the references were reviewed and formatted according to the journal guidelines. We also checked all the manuscript searching for double spaces and typos.
Reviewer 3 Report
The paper of Fonseca et al. describes the identification of functional candidate genes associated with fertility status in beef cattle. Taking advantage from RNA-sequencing data, the authors made use of two similar datasets in order to produce a co-expression gene network (via a meta-analysis approach) that led to the identification of functional modules, hub-genes and genetic variants putatively affecting fertility status. Overall, the paper is interesting and well written. The bioinformatic pipeline technically sounds, is well presented and described. I have just some minor comments:
- line 110 and other lines: "p" of p-value should be in italics
- line 111: add references papers?
- line 157: how many pathways have you tested? Size of background? Did you use FDR or Bonferroni correction?
- line 220: Tale S2: Are results coming from VEP analysis? It is not stated in the main text.
- Figure S2: Better to highlight the excluded animals.
- Line 207-209 should be removed?
- Line 407: 101?
- Discussion can be a bit shortened.
Author Response
Reviewer 3
The paper of Fonseca et al. describes the identification of functional candidate genes associated with fertility status in beef cattle. Taking advantage from RNA-sequencing data, the authors made use of two similar datasets in order to produce a co-expression gene network (via a meta-analysis approach) that led to the identification of functional modules, hub-genes and genetic variants putatively affecting fertility status. Overall, the paper is interesting and well written. The bioinformatic pipeline technically sounds, is well presented and described. I have just some minor comments:
Thank you for all the valuable comments and suggestions. We addressed all the points raised during the review. The modifications are highlighted in yellow in the current version for he manuscript. We are addressed the individual questions bellow.
line 110 and other lines: "p" of p-value should be in italics
Answer: Thank you for the comment, we changed all the p of p-values by italic in this version of the manuscript.
line 111: add references papers?
Answer: Thank you for the comment. We added the reference for this sentence in the current version of the manuscript.
Love, M. I., Huber, W., Anders, S. Moderated estimation of fold change and dispersion for RNA-seq data with DESeq2. Genome biology 2014, 15(12), 550.
line 157: how many pathways have you tested? Size of background? Did you use FDR or Bonferroni correction?
Answer: Thank you for the comment. All the tested pathways are available on Supplementary Table 5. In total, 271 canonical metabolic pathways were tested. As background we used the annotated genes in the human, mouse and rat genome available on IPA. In the present study, we didn’t apply a correction for multiple testing for the pathway enrichment analysis. We didn’t choose the candidate modules or functional candidate genes based on the enriched pathways. We used this information only as an additional source of functional annotation. Therefore, even the non-enriched pathways would add some additional information and help to understand the function of the genes within the candidate modules.
line 220: Tale S2: Are results coming from VEP analysis? It is not stated in the main text.
Answer: Thank you for the comment. We included the information in the current version of the manuscript (Line 219).
Figure S2: Better to highlight the excluded animals.
Answer: Thank you for the comment. We highlighted the excluded animals IDs on lines 225-226.
Line 207-209 should be removed?
Answer: Thank you for the comment. We would like to keep the workflow summarizing the pipeline applied in the current study. As the analysis are composed by several step, in our opinion, the workflow is important to make easer to have a complete overview of all the analyses performed.
Line 407: 101?
Answer: Thank you for the comment. The respective reference was adjusted.
Discussion can be a bit shortened.
Answer: Thank you for the suggestion. Taking into account the suggestion of the other reviewers, we were not able to short the discussion section. We are aware that the discussion section is long. However, there are several concerns about the differences between the datasets and multiple differentially co-expressed modules and its respective functional candidate genes that must be discussed, which justify a longer discussion section.